# Hippocampal Sclerosis in Pilocarpine Epilepsy: Survival of Peptide-Containing Neurons and Learning and Memory Disturbances in the Adult NMRI Strain Mouse

**DOI:** 10.3390/ijms23010204

**Published:** 2021-12-24

**Authors:** Adrienne Mátyás, Emőke Borbély, András Mihály

**Affiliations:** 1Department of Anatomy, Albert Szent-Györgyi Faculty of Medicine, University of Szeged, Kossuth L. sgt. 38, H-6724 Szeged, Hungary; matyas.adrienne@med.u-szeged.hu; 2Department of Medical Chemistry, University of Szeged, Dóm tér. 8, H-6720 Szeged, Hungary; emokeborbely@gmail.com; 3Professional Pedagogical Service of Csongrád-Csanád County, Űrhajós u. 4, H-6723 Szeged, Hungary

**Keywords:** hippocampus, epilepsy, pilocarpine, mouse, neuropeptide-Y, parvalbumin, calretinin, learning

## Abstract

The present experiments reveal the alterations of the hippocampal neuronal populations in chronic epilepsy. The mice were injected with a single dose of pilocarpine. They had status epilepticus and spontaneously recurrent motor seizures. Three months after pilocarpine treatment, the animals were investigated with the Barnes maze to determine their learning and memory capabilities. Their hippocampi were analyzed 2 weeks later (at 3.5 months) with standard immunohistochemical methods and cell counting. Every animal displayed hippocampal sclerosis. The neuronal loss was evaluated with neuronal-N immunostaining, and the activation of the microglia was measured with Iba1 immunohistochemistry. The neuropeptide Y, parvalbumin, and calretinin immunoreactive structures were qualitatively and quantitatively analyzed in the hippocampal formation. The results were compared statistically to the results of the control mice. We detected neuronal loss and strongly activated microglia populations. Neuropeptide Y was significantly upregulated in the sprouting axons. The number of parvalbumin- and calretinin-containing interneurons decreased significantly in the Ammon’s horn and dentate gyrus. The epileptic animals displayed significantly worse learning and memory functions. We concluded that degeneration of the principal neurons, a numerical decrease of PV-containing GABAergic neurons, and strong peptidergic axonal sprouting were responsible for the loss of the hippocampal learning and memory functions.

## 1. Introduction

The pilocarpine (PILO) rodent’s epilepsy is one of the widely used animal epilepsy models [1]. Pilocarpine is the lipophilic agonist of the muscarinic acetylcholine receptors [1]. The convulsive effect is initiated through the muscarinic receptors located on the hippocampal principal neurons [2]. The muscarinic excitation causes the release of excess glutamate, which will be responsible for the status epilepticus (SE) through NMDA receptor activation [1,3]. The generalized motor convulsions cause astrocytic swelling, neuronal damage, and the opening of the blood–brain barrier (BBB) [4,5,6]. These primary neuropathological events will initiate subsequent neuroinflammatory cascades. The opening of the BBB gives way to blood-borne immune cells which enter the brain parenchyma [7,8]. The local microglia activation releases citokines, which further stimulate neuroinflammation (e.g., the sprouting of microvessels and migration of macrophages). Therefore, the consequence of the SE will be a slowly commencing neurodegeneration, which is also marked by spectacular alterations of the neuronal networks, including neuronal cell loss in CA1 and in CA3, the dispersion of dentate granule cells, and the sprouting of mossy axons in the dentate gyrus [1,9,10,11]. Similar to human temporal lobe epilepsy (TLE), chronic pilocarpine epilepsy in rodents is characterized by spontaneous recurrent seizures (SRS) [1] and ongoing neuronal degeneration, which ultimately results in hippocampal sclerosis (HS) [12]. The SRS in rodents are involuntary motor episodes which can vary from brief, nearly undetectable muscle twitches to long periods of vigorous shaking [1]. The symptomatic convulsions are caused by hyperactivity of the cerebrocortical neuronal networks. This hyperactivity can be measured by electroencephalography (EEG) [1] and magnetic resonance imaging (MRI) [13] of the experimental animals. The extent of neuronal degeneration is regularly proportional to the severity of the SRS [1]. These animal epilepsies with long-lasting SRS cause permanent brain damage (e.g., shrinkage of the hippocampus and amygdala, similar to TLE patients) [14]. These convulsing animals are often used for pharmacological studies in order to develop antiepileptic therapeutic strategies [6,11,15,16].

Pilocarpine-treated (PT) animals display SRS for months [1,17]. Histological analysis revealed that pilocarpine-induced SRS causes variable loss of the pyramidal and mossy cells, strong mossy fiber sprouting, and reorganization of the ionotropic glutamate receptor subunits in the hippocampus [1,9,18,19,20]. Parallel studies in patients and animal models indicated that the appearance of hypersynchronous neuronal discharges is responsible for the generation of spontaneous motor seizures [21]. The persistent hyperactivity of dentate interneurons and neuronal degeneration result in the loss of balance between the excitatory and inhibitory neuronal circuits in the hippocampal formation (HF) [21,22,23].

Although the HS is a term for the description of the neuropathological findings in human TLE [12], similar neuropathological changes occurred in the animal models of TLE: neuronal (mainly pyramidal cell) loss in CA1-4 [10], the dispersion of granule cells in the dentate gyrus (DG) [11], and axonal sprouting in the mossy fiber system [17]. The complex neuropathological phenomena of the HS coincide with significant cognitive disturbances in human patients [24].

The aims of the present study are as follows:(1)We aimed to analyze the neuropathology of HS in the hippocampus of NMRI mice surviving 3.5 months after the initial SE. The NMRI mouse strain is frequently used in experimental epilepsy studies [11,17,18]. Therefore, new data on HS in NMRI mice could have been useful.(2)We wanted to investigate the survival of two interneuron populations (PV- and CR-containing interneurons) in HS, because the interneurons are important not only in normal but also in pathological conditions [18,19,21].(3)We aimed to study the effects of HS on the spatial learning and memory processes in the NMRI mouse strain.

We detected the neuronal loss with Neuronal-N (NeuN) labeling [25] and the axonal sprouting with neuropeptide Y (NPY) immunostaining, which similarly to Timm’s sulphide silver method [26] are accepted labeling methods in the pathology of HS [12,17]. We concluded that HS caused a significant decrease in the number of hippocampal principal- and peptide-containing neurons and a concomitant microglia proliferation in the damaged hippocampal layers.

## 2. Results

The PILO-injected animals developed acute motor symptoms including tremors, automatisms, wild running, and tonic-clonic seizures 10–15 min after the PILO injections. The symptoms lasted for 90 min or more, and therefore we termed this SE. The animals received intraperitoneal diazepam injections 90 min after the beginning of the motor symptoms. The mortality rate of the PILO-treated (PT) animals was 75% in our experiments during the 3.5-month observation period, which was slightly higher than that stated in the literature [1]. Following the SE, the animals went through a latent period of 1–2 weeks, they then presented daily short motor seizures (SRS), which consisted of limb tremors, short wild running, and frequent automatisms lasting for 1–5 min. During observation, the SRS symptoms of the seizing animals were compared to the healthy controls.

### 2.1. Neuronal Loss in the Hippocampus 3.5 Months after the SE

We measured the optical density of the NeuN immunostaining with the area of interest (AOI) being the entire HF: the subiculum, the Ammon’s horn (CA), and the DG. The subiculum contained neurons in every PT animal, although we did not evaluate separately the NeuN staining density in this sector. The NeuN-stained cells disappeared mainly from the pyramidal layer of the CA1, CA2, and CA3 sectors of the Ammon’s horn in animals PT1, PT3, PT4, PT6, PT7, PT8, and PT9 (Table 1 and Figure 1). In these animals, scattered NeuN-stained cells were present only in the strata oriens, radiatum, and lacunosum moleculare (Figure 1A). These cells were probably interneurons. In animal PT2, the pyramidal layer was seemingly undamaged in CA1, but the CA2 neurons in the pyramidal layer were completely degenerated. In animal PT8, the pyramidal layer in CA1 and CA3 displayed patchy neuronal losses; neurons of the stratum pyramidale were missing in 100–200-µm long periods. The granule cells of the dentate gyrus did not display spectacular loss, although in PT6 and PT7, the lateral tip of the upper blade of the granule cell layer was thinner than normal, and small NeuN-stained cells were scattered, suggesting a decrease in the granule cell number in this area. On the other hand, the thickening of the rest of the DG granular layer was observed in the animals with sclerosed hippocampi, suggesting the dispersion of the granule cells [11,12]. The granule cell layer was not counted separately. Densitometric analysis of the NeuN-like staining in the hippocampus (subiculum, CA1, CA2, and CA3; Figure 2A) and in the DG (hilum, granule cell layer, and molecular layer; Figure 2B) proved a significant loss of neurons in PT animals (Figure 1 and Figure 2). The statistical analysis of the NeuN staining densities displayed approximately a 50% loss of immunostaining in the Ammon’s horn and approximately a 30% staining loss in the DG (Figure 2A,B). The localization of the neuronal degeneration in the individual PT animals is displayed in Table 1. The qualitative analysis of the immunohistochemical sections proved that every PT animal displayed HS. According to the significant neuronal loss, the measured cross-sectional area of the hippocampal formation (HF) decreased significantly in the epileptic animals, indicating shrinkage of the HF (Figure 3A).

### 2.2. Microglial Activation and Proliferation 3.5 Months after the SE in the HF

The Iba1-immunoreactive microglia cells were found to be scattered in the normal HF (Figure 1B). The cell bodies were small, and the processes were slender and relatively short. The epileptic HF contained a significantly higher number of immunostained microglia cells, as shown by the significant increase in Iba1 staining density (Figure 1D). The cells were larger, and the processes were thicker, longer and more numerous than those of the controls [27]. A comparison of the NeuN-stained hippocampi and the Iba1-stained ones resulted in a negative “match”. Those hippocampal areas which lost their neurons contained a high number of enlarged microglia cells in or close to the pyramidal layer (Figure 1C,D). The overall staining density of the microglia increased significantly in the HF of the PT mice, indicating an increase in the number of microglia cells (Figure 3B) [27].

### 2.3. Sprouting of Mossy Fibers 3.5 Months after the SE

The control mice displayed scattered NPY immunoreactive neuronal somata in the hippocampus in the strata oriens, pyramidale, and radiatum. The DG has NPY-positive cell bodies in the hilum and close to the granule cell layer, with processes sometimes extending between the granule cells into the molecular layer [17]. We did not count the NPY-stained cells in our experiments. The dendritic processes of the cells also contained NPY immunoreactivity mostly in strata oriens and lacunosum-moleculare of the CA3 sector between the granule cells and moderately in the molecular layer of the DG (Figure 4A). Axonal staining was not visible in the controls (Figure 4A). The PT mice presented strong NPY staining of the neuropil in the entire hilum and in the stratum lucidum and stratum oriens of CA3 (Figure 4B). Strong NPY-like staining was present in the inner molecular layer (IML) of the DG. These NPY-IR elements were probably contacting the primary dendrites of the granule cells (Figure 4C). The localization of the NPY staining was very similar to the Timm reactivity of these areas described in the literature [17,18]. Therefore, we think that these NPY-containing elements are the sprouting axons [17,18]. The NPY staining of these hilar and CA3 areas was found in every PT animal, irrespective of the neuronal damage. Even if there were no nerve cells in CA3, the NPY-immunostained network was present in the area corresponding to the stratum lucidum of CA3 (Figure 4C). The axonal sprouting was not quantified because it appeared as an all-or-nothing phenomenon, with nothing in the controls and strong NPY staining in every PT animal.

### 2.4. Parvalbumin Containing Hippocampal Interneurons 3.5 Months after the SE

The PV-containing neurons are scattered in the HF. Most of them are located in the stratum pyramidale, where a dense PV-immunoreactive (PV-IR) fiber network is seen (Figure 5A) [22]. Scattered PV-IR neurons are present in the strata oriens and radiatum (Figure 5A). In the DG, PV-IR cells are located in the stratum granulosum and hilum (Figure 5A). Because of the different degrees of neuronal damage, we did not evaluate the regions of the hippocampal formation separately. In HS, it was not possible to determine the location of the CA sectors precisely. We counted the overall number of PV-IR cells in the entire area of the HF. A significant decrease in the number of PV-IR cells was found. Some of the remaining PV-IR cells were found in the hilum, in the granular layer of the DG, in the stratum oriens and stratum radiatum of the Ammon’s horn, and some were in the subiculum (Figure 5B,C). These remaining PV-IR cells displayed strong immunoreactivity (Figure 6A). We detected PV-stained astrocyte-like cells in the stratum lucidum of CA3 and in the molecular layer and hilum of the DG in the sclerosed hippocampi (Figure 6B). The significant decrease in PV cells was shown by the independent samples *t*-test, the Wilcoxon signed rank test, and by two-way ANOVA. The descriptive statistics proved there were 33 cells (mean value) in the controls and 16 cells (mean value) in the PT animals in 1 mm^2^ of the HF. This indicated an approximately 50% decrease in the number of PV neurons in the HF of the PT animals compared with the controls (Figure 7A).

### 2.5. Alterations in the Number of Calretinin Immunostained Cells

The CR-containing interneurons were scattered in the normal hippocampi. They appeared in the CA fields, mainly in the stratum lacunosum-moleculare (close to the hippocampal fissure), stratum radiatum, oriens, and pyramidale. They were present in the hilum of the DG as well, where they were located beneath the granule cell layer, in the center of the hilum, and rarely between the granule cells. Their dendrites and axons extended into the molecular layer and also formed a plexus in the hilum beneath the granule cells. Strong CR immunoreactivity (CR-IR) was visible in the IML of the DG. This CR staining seemingly originated from thin CR-IR processes and was present in the controls and PT animals (Figure 8A,B). The number of CR-IR cells decreased significantly in the HF of the PT animals, although the decrease was less pronounced than that of the PV cells (Figure 7B). We detected a mean CR-IR cell number of 19 in the controls and a mean CR-IR cell number of 15 in the PT animals. This was an approximately 20% decrease in the number of CR-IR neurons in the PT animals compared with the controls (Figure 7B). In one mouse (PT1), the dentate granule cells expressed strong CR-IR (Figure 8A,C). The CR-IR granule cells were equally numerous in the ventral and dorsal blades of the granule cell layer. Morphologically, they appeared as average granule cells which contained CR in their somata and proximal dendrites (Figure 8C). Other epileptic animals (PT2–PT9) displayed CR staining of thin varicose fibers in the hilum and IML close to the granule cells (Figure 8B). Large multipolar neurons in the hilum (possibly mossy cells) were stained strongly with the CR antibody both in the control and PT animals, according to the literature descriptions [28]. Higher magnification always revealed several thin, beaded CR-IR neuronal processes in the entire hilum and in the entire molecular layer (Figure 8B).

### 2.6. The Impairment of Spatial Learning 3 Months after the SE

The effects of PILO-evoked SE and the chronic SRS episodes on the spatial learning and memory functions were tested in a Barnes maze 3 months after PILO treatment. The test was based on two daily spatial navigation experiments, acquisition of the spatial information in the maze, and the successful retrieval of the spatial memory on the fourth day of testing (probe trial). In the Barnes maze, the control animals found the escape hole fast on every trial occasion (Table 2; Figure 9). The control animals were successful in the day 4 retrieval phase as well. The descriptive statistics showed a mean latency of 36.10 s (SD = 25.48), while the PT mice produced a mean latency of 139.8 s (SD = 113.1). Using the Mann–Whitney test, the difference between the latencies of the control and PT mice during the probe trial was significant (*p* = 0.0295). The latencies of day 1–3 were not different statistically (Table 2). The spatial learning curves demonstrated that the chronic seizures induced a marked decrease in learning as well as in spatial memory performance (Figure 9). The effective learning process was demonstrated in the controls by the slight daily decrease of the latencies (Figure 9). The PT mice did not show a decrease in latencies. Instead, the latencies increased slightly every day (Figure 9; Table 2). Time-related XY analysis of the latencies of the controls and the PT animals proved that in the PT group, the variance of the means of the daily latencies was significantly different (*p* = 0.0438) (Figure 9).

The following animals were the worst performers in the Barnes maze: PT1, PT4, PT5, and PT8 (see also Table 1 for comparison). PT1 and PT4 had completely degenerated CA regions (CA1, CA2, and CA3). PT5 lost the CA3 region completely and also had cell loss in the hilum. PT8 had patchy cell losses in the entire CA1 and CA3 regions.

## 3. Discussion

PILO seizures in rodents serve as an accepted model experiment for human TLE [1,9]. Following systemic PILO administration, generalized motor convulsions develop which last for more than 1 h. Most animals present SE, and those which survive enter into a latent phase without symptoms, after which the animals develop SRS [1,9]. The brain damage is proportional to the length of the SE and the frequency of the SRS [1,9,19]. We have to note that the acute SE is already damaging the brain via astrocytic swelling, neuronal damage, and cell death, and microglial activation and BBB damage follow the SE during the first week of the survival [27,29,30]. These complex neuropathological events initiate the ongoing neurodegeneration, which ends up as the HS months after PILO injection [1,9,17]. Here, we described the neuropathological sequelae of 9 NMRI strain mice, which presented SRS for 3.5 months and developed a characteristic HS.

### 3.1. Neuronal Loss in Epileptic Hippocampi

Neuronal damage is a consequence of acute SE [4,23,31,32]. Hypoxic conditions develop in the brain parenchyma during the SE, where in spite of the increase in the regional cerebral blood flow, neurons are not getting enough oxygen [32]. Due to the extracellular accumulation of glutamate, potassium, hydrogen carbonate, and ammonia, the astrocytes swell, obliterate the extracellular space, and hinder the diffusion of nutrients [31,32]. Neurons are deprived of nutrients (e.g., glucose and amino acids), and this condition together with the hypoxia leads to cell damage and necrosis [31,32]. These processes, which lead to acute neuronal cell death, are well documented in the literature [31,32,33]. The neuronal degeneration in the hippocampus was clearly demonstrated by our present experiments. The PT mice demonstrated the doctrine of the graded sensitivity of the hippocampal subfields [33]. CA1 and CA3 were the areas where the pyramidal layer mostly suffered. The granule cells of the DG were spared, although some neuronal loss was detectable in the DG, too. According to the neuronal loss, the cross-sectional area of the HF decreased significantly, as measured in our experiments. This shrinkage of the HF is regularly observed in TLE in human patients [34].

### 3.2. Microglial Activation in the Epileptic Mice

Microglia cells are activated rapidly following pathological changes of the brain in epilepsy [27,30]. The widespread expression of neurotransmitter receptors on microglia cells renders the cells sensitive to neuronal hyperactivity [35]. These receptors are initiating the transformation of the microglia cell and also regulate their citokine secretion [35]. Microglia can promote neurodegeneration [36] but can also be a protective device by detecting the damage and also exerting a protecting role in neurodegeneration [37]. Our experiments clearly pointed to this role. The activated microglia cells were seen in those layers which suffered a massive neuronal loss (e.g., the stratum pyramidale of CA1). This phenomenon points to the possibility that disease-associated microglia (DAM) occupied the sclerotic hippocampal subfields in order to maintain a kind of homeostasis after the death of the neurons [37]. However, this is a hypothesis which has to be validated through the immunohistochemical detection of markers of the DAM [37]. Our previous experiments proved that the microglial response in PILO epileptic mice was very strong and could be inhibited by the synthetic kynurenic acid analog SZR104 [27]. The significance of the inhibition is not yet clear, but we interpreted the inhibitory action of SZR104 in terms of the aryl hydrocarbon receptor [27].

### 3.3. Sprouting of Mossy Fibers in the Epileptic Mice

The sprouting of mossy axons in epilepsy is well documented not only in experimental animals [17,18,26,38] but also in the human brain [12,39]. The main layers targeted by the sprouting axons are the molecular layer of the DG, the hilum of the DG, and the stratum lucidum of CA3 [17,18,26,38,39]. Other locations where sprouted axons terminate are the infrapyramidal sublayer of CA3 stratum oriens, the granule cell layer, and the IML of the DG [39]. We detected the sprouting with NPY-immunohistochemistry and observed the ectopic axons even in case of complete loss of the principal neurons in the area corresponding to the stratum lucidum of CA3. This observation indicates that these sprouted axons persisted without postsynaptic targets. They probably released NPY, and the released peptide exerted its anti-epileptic effect through diffusion to the DG. The increase in NPY expression was detected in the granule cells by immunohistochemistry [40] and in situ hybridization [41], proving the alteration of the granule cell phenotype in epilepsy [41]. The NPY causes depression of the granule cell epileptiform activity [42]. The sprouting plays an important role in the reorganization of the hippocampus, and it is a standard feature of the human HS in TLE [12,39].

### 3.4. The Decrease in the Number of the PV-Containing Interneurons

The PV-containing interneurons of the hippocampal formation are located mainly in the pyramidal and granule cell layers and the hilum [43]. They are mostly GABAergic basket cells and axo-axonic cells exerting inhibition on cell bodies, proximal dendrites, and axon initial segments [43]. The number of these neurons decreased significantly in the sclerosed hippocampi in our experiments, indicating the decrease of inhibition in the pyramidal and granular layers as documented by the literature data [21,22,43]. Similar observations were described in experimental rats suffering absence epilepsy [44] and kindling epilepsy [45].

Degeneration in the pyramidal layer swept out not only the principal cells but also the PV-IR GABAergic cells. We encountered only very few PV-IR cell bodies (3–4 in total), located close to the alveus and in the subiculum. The estimated decrease in the PV-IR cell population was about 50% in our experiments. The literature data support this observation [22,34,46]. The presence of scattered PV-IR astrocytes in the sclerosed regions is a new finding. The upregulation of PV in reactive astrocytes and ependymal cells was described in vivo [47] and in vitro [48]. The upregulation was caused by brain injury [47] and virus infection [48]. In these experiments, PV upregulation was discussed in connection with changes in cell metabolism and proliferation signaling [47,48]. In our experiments, the alterations of the microenvironment due to neurodegeneration and axonal sprouting could be the explanation for astrocytic PV expression. However, this issue has to be examined in further experiments with immunostainings using PV antibodies and specific astrocyte markers.

### 3.5. Alterations of CR Immunoreactivity in the Hippocampal Formation

We described the decrease in the number of CR-IR neurons in the epileptic hippocampus. This decrease was approximately 20% in terms of the total number of cells/mm^2^. The large SD of the mean (Figure 6B) could be the consequence of the expression of CR in the dentate granule cells in one mouse (PT1). The expression of CR in the granule cells could be explained by the phenotype changes of these neurons in epilepsy [41]. The mossy cells also express CR in normal and epileptic mice, and their number decreases in epileptic animals as shown in the literature [28]. The decrease in other hippocampal CR-IR cells is probably connected to HS. The degeneration of the principal neurons (which are synaptic targets of the interneurons) is accompanied by the degeneration of the interneurons, such as in the case of PV-IR cells. These observations about the decrease are supported by the literature data [45,49]. Similarly, the decrease in DG CR-IR cells was seen in the dorsal hippocampus in kindling rat experiments [45]. On the other hand, a significant increase in the number of CR-IR Cajal–Retzius cells was detected in epileptic human brain samples [50,51]. The sprouting of CR-IR nerve fibers of the DG was also noted in human HS [12]. Our present observations on the abundance of thin CR-IR fibers in the DG in epileptic mice supports the possible participation of CR neurons in sprouting and reorganization of the DG in HS. However, this ambiguity of the CR neurons in epilepsy has to be further investigated.

### 3.6. Learning and Memory Deficiency in Epileptic Mice

Our experiments proved that the spatial learning processes were slower during the acquisition trials, and the retrieval of spatial memory was significantly worse during the final probe process in the epileptic versus control mice. This is in accordance with the literature data [52]. Similar (although less dramatic) impairment of spatial learning was seen after kainic acid lesions of the CA1 sector of the dorsal hippocampus in mice [52]. The non-epileptic experimental lesions of the dorsal and ventral hippocampus are also associated with impairments in working memory and reference memory retrieval [53]. A similar spatial memory disturbance occurred following the lesion of the lateral entorhinal cortex in our previous experiments [54]. The entorhinal cortex and the hippocampus are important in the spatial learning of rodents [54]. The HS destroys not only the principal neurons but also several interneurons, and therefore the reciprocal neuronal connections between the neocortex, entorhinal, and hippocampal cortices will be interrupted. We detected HS in the histological sections of the dorsal hippocampus in the present experiments. The neuronal loss in the dorsal HF could well explain the spatial memory impairment seen in the Barnes maze experiments [52]. Similar results were reported in epileptic C57BL/6 mice, where the hippocampal neuronal loss was correlated with a significant increase in escape latency in the Morris water maze tests [55]. The neuronal loss in CA1, CA3, and the hilum of the DG were evaluated on thionin-stained sections of the control and PT animals 9 months after PILO administration, and significant neuronal loss was found, which was correlated with the Morris water maze results [55]. We also could select the PT animals with the worst learning and memory performance. However, further experiments are needed to exactly correlate the degree and localization of neuronal loss, gliosis, and learning and memory performance. The correlation of HS and memory loss in our experiments has clinical significance, as the damage to the hippocampus in human TLE has also been correlated with visuospatial memory loss in adult patients [56].

## 4. Materials and Methods

### 4.1. Experimental Protocol and Pharmacological Treatment

Adult, male NMRI strain mice (n = 50, average body weight (bwt) of 35 g) were kept in a temperature-controlled room (24 °C ± 1 °C, humidity: 22% ± 2%) under standard a dark/light cycle (12/12 h) with food and water ad libitum. All experimental procedures and handling and housing of the animals were conducted according to the European Union Directive (2010/63/EU4) and according to the Hungarian Animal Act. Specific approval of care and use of animals was obtained in advance from the Faculty Ethical Committee on Animal Experiments (University of Szeged; 1-74-1/2017. MÁB) and from the Government Office (Department of Food and Animal Health) of Csongrád County (CS/101/3347-2/2018).

The animals were divided into two experimental groups: one group was injected with PILO (40 animals), and the other group (controls) received the solvent of PILO (10 animals). During the experiment and after the treatments, the animals were individually labeled on the tail skin and observed daily in their cages in order to note the spontaneous behavioral seizures which they experienced. The daily observation time was 1 h over a 9–10 a.m. period. The survival period lasted for 3.5 months following PILO treatment.

The experiments were performed in the morning. The PT animals (n = 40; 28–36.1 g bwt) received a single 195 mg/kg dose of pilocarpine hydrochloride (Sigma-Aldrich, St. Louis, MO, USA) dissolved in a 0.9% NaCl solution (physiological saline) injected intraperitoneally (i.p.). PILO precipitated motor seizures, and 90 min after the first motor symptoms of the epileptic seizure, 10 mg/kg diazepam was injected i.p. (Seduxen, Gedeon Richter, Budapest, Hungary) in order to terminate the motor convulsions. Post-seizure treatment included continuous observation (every day until 7 p.m., starting in the morning at 5 a.m. and 9 a.m., then at noon, and finally at 3 p.m.) and intradermal and i.p. injections of Krebs-Ringer bicarbonate buffer solution (bicarbonate buffered isotonic saline containing magnesium, potassium, sodium, and glucose; Merck KGaA, Darmstadt, Germany) during the first 24 h. Nine animals survived the 3 months, and these animals (together with nine controls) were used in the Barnes maze experiments, and then 2 weeks later, their brains were investigated with immunohistochemical techniques (Table 1).

The control animals (n = 10) were injected i.p. with the same volume of physiological saline, the solvent of PILO. These animals displayed normal behavior and did not receive diazepam or Krebs-Ringer solution injections. Nine of these animals were also tested 3 months later with the Barnes maze, and 2 weeks later, their brains were used in immunohistochemistry experiments (controls).

### 4.2. Tissue Preparation

Three and a half months after the injections and 2 weeks after the Barnes maze experiments, the animals were sacrificed. The mice (9 PT and 9 control animals) were deeply anesthetized with halothane (Sigma-Aldrich, St. Louis, MO, USA) and perfused transcardially with 30 mL of physiological saline followed by 30 mL cold 4% paraformaldehyde in a 0.1-M phosphate buffer (PB, pH 7.4). The brains were removed and postfixed in the same solution for 5 h and then cryoprotected overnight in 30% sucrose in a 0.1-M phosphate buffer (PB, pH 7.4) at 4 °C. Coronal brain tissue blocks were cut using an acrylic brain slicer matrix (Zivic Instruments, Pittsburgh, PA, USA). Coronal plane sections were cut at a 25-μm thickness from the hippocampus (between coronal levels Bregma −1.06 mm and Bregma −2.30 mm [57]) on a freezing microtome, and the sections were stored at 4 °C in a PB solution containing 0.1% NaN_3_ until further processing.

### 4.3. Immunocytochemistry

Free-floating sections were used. Endogenous peroxidase activity was blocked with 3% H_2_O_2_ for 15 min. Nonspecific binding sites were blocked with 20% normal pig serum (NPS; diluted: 1/10), and tissue permeability was enhanced by using 1% Triton X-100 in the blocking NPS solution. The sections were incubated overnight at room temperature in primary antibody solution with the following primary antibodies: rabbit anti-neuropeptide Y (NPY; 1/10,000, Abcam, Cambridge, UK); mouse anti-Neuronal N (NeuN; 1/8000, Chemicon, Temecula, CA, USA); rabbit anti-Iba1 (Iba; 1/8000, FUJIFILM Wako Chemicals Europe GmbH, Neuss, Germany); goat anti-calretinin (CR; 1/20,000, Millipore, Temecula, CA, USA); and mouse anti-parvalbumin (PV; 1/40,000, Sigma-Aldrich, St. Louis, MO, USA). Biotinylated secondary antibodies (1:400 dilution, Jackson Immuno Research, West Grove, PA, USA) were used for 1 h at room temperature, and the signal was detected with peroxidase-labeled streptavidin (1/6000, Rockland Immunochemicals Incorporation, Limerick, PA, USA) for 1 h at room temperature. All incubations were performed in plastic vials with continuous agitation. The sites of the immunoreaction were visualized with diaminobenzidine tetrahydrochloride (DAB) + H_2_O_2_ in the absence (30 min) or presence of 0.3% nickel sulphate (15 min reaction time, used only with the NPY antibody). The sections were mounted on microscope slides, air-dried, dehydrated, and coverslipped with DPX (Merck KGaA, Darmstadt, Germany). Chemicals other than antibodies were purchased from Sigma-Aldrich.

### 4.4. Morphometry and Evaluation of the Data

The stained sections were scanned with a slide scanner (Mirax Midi, 3DHistech Ltd., Budapest, Hungary) equipped with a Pannoramic Viewer 1.15.4, CaseViewer 2.1 program, and a QuantCenter HistoQuant module (3DHISTECH Ltd., Budapest, Hungary). Using the digital images, the AOIs were manually outlined. The NeuN and Iba1 immunostainings were quantified with the DensitoQuant application of the QuantCenter (3DHISTECH Ltd., Budapest, Hungary). The application identifies the positive stain based on an automatic color separation method, through which individual positive pixels are counted and classified based on intensity and threshold ranges. In this way, a staining density score is calculated by the software, based on the proportion of positive and negative pixels. The staining density scores are depicted in the diagrams as percentage values following statistical testing. In the case of NeuN and Iba1 stainings, the density score measurements from every control and every PT animal were used (9-9 measurements). The 9 staining density values were analyzed statistically, and the graphs were constructed with the help of GraphPad Prism version 9.2.0 software (GraphPad Software, LLC, San Diego, CA, USA). The area of the hippocampal formation was measured by the QuantCenter after manually outlining the HF on the histological section. In these measurements, hippocampi from each animal (controls and PT animals) were measured, and the single area measurements (n = 9 control, n = 9 PT) were treated statistically in order to illustrate the shrinkage of the HF. The NeuN and Iba1 staining densities and hippocampal area measurements were analyzed with both parametric (paired *t*-test) and non-parametric (Wilcoxon and Mann–Whitney) tests with the help of GraphPad Prism version 9.2.0 software. Every statistical test resulted in significant *p* values (see Figure 2 and Figure 3).

The PV- and CR-immunostained cells were counted manually using the digitized images. The AOIs were manually outlined, representing the HF. The immunostained cells were spotted on a screen with 150× magnification. Only those cells which displayed sharp contours and processes were counted. The number of the counted cells was then corrected to a 1-mm^2^ area of the HF. The data of the PV and CR neuron counts were analyzed with parametric (paired *t*-test) and non-parametric (Wilcoxon and Mann–Whitney) tests (GraphPad Prism 9.2.0; GraphPad Software, LLC, San Diego, CA, USA; see Figure 6).

### 4.5. Spatial Learning Test

Three months after the injections, the animals (control n = 9; PT, n = 9) were tested in a hippocampus-related learning paradigm, the Barnes maze, to measure the spatial learning and memory capabilities [57]. The maze is based on the aversion of rodents to open spaces, which motivates them to search for shelter in the escape box. The experiments were carried out in a well-lit testing room with a holding room located in the vicinity. The Barnes maze apparatus consists of a circular, black surface arena (plate diameter: 100 cm; thickness: 1 cm; the plate is located 90 cm above the floor) with 18 circular holes around its circumference. Under one of the circular holes, there is a black escape box where the animal can hide [58]. The arena is located in a small experimental room with relatively large visual cues, such as colored shapes attached to the wall to enable the mice to learn the locations of the holes. The arena surface is brightly lit by an overhead light source. The “escape box” can be reached through the corresponding hole on the tabletop. We did not change the location of the escape box during the experiment. The animal was placed in a non-transparent starting cylinder in the center of the arena for 10 s. After lifting the cylinder, the mice were given 5 min to explore the holes and 15 s to be in the box. The animals that failed to find the escape box were gently guided into the hole and were allowed to stay there for 15 s. Following the lifting of the cylinder, the time needed for finding the escape box was measured with a chronometer by an investigator (E.B.) who was not aware of the animal treatment. No video tracking was used, and only the latencies of finding the escape box were measured. All animals performed 2 trials per day with a constant intertrial interval of 4 h for 3 consecutive days. The plate and the escape box were cleaned with 70% ethanol after each session and for each animal in order to eliminate olfactory cues. The performance was evaluated through the measuring the time of the latency to find the escape box. The latency measurements were analyzed with the Mann–Whitney test (Table 2) and two-way ANOVA (GraphPad Prism, version 9.2.0 software, GraphPad Software, LLC, San Diego, CA, USA). The Mann–Whitney test was used to compare the daily measurements. We compared the average latencies on days 1–4 in the controls and PT animals (column analysis [58]; see Table 2). The measurements were also analyzed in relation to time. The daily latencies were plotted as Y values (dependent variables), while the days were plotted as X values (independent variables). Two-way ANOVA [58] was used for statistical testing. Our null hypothesis was that the variance of the means of the latencies in the two groups was not different [59]. The results of the statistical analyses were reported according to Khakshooy and Chiapelli [59].

## 5. Conclusions

The HS in epileptic NMRI mice was characterized by neuronal loss, microgliosis, and axonal sprouting. The neuronal loss in the Ammon’s horn was approximately 50% compared with the controls. The neuronal loss in the DG was less, being approximately 30% compared with the controls.A large population (approximately 50%) of the hippocampal PV-containing GABAergic neurons was lost in HS. The loss of PV interneurons possibly contributed to the decay in learning and memory performance of the PT animals.The degeneration of hippocampal CR-containing interneurons was less pronounced, as approximately 20% of the CR-containing cells degenerated in HS.Axonal sprouting involved large areas of the epileptic hippocampal neuropil. The hilum and IML of the DG, the stratum lucidum, and stratum oriens of the CA3 were heavily innervated by axons containing the NPY transmitter peptide.The strong increase in NPY expression proved that in HS, the peptidergic component of the hippocampal neurotransmission survived better than the glutamate system. This possibly could lead to alterations of the hippocampal synaptic transmission. Instead of fast glutamate signaling, the slow, metabotropic peptidergic signaling prevailed. The slowdown of signaling could also contribute to the worsening of learning and memory functions.We summarized the neurochemical transformation of the sclerosed hippocampus in Figure 10.

## Figures and Tables

**Figure 1 ijms-23-00204-f001:**
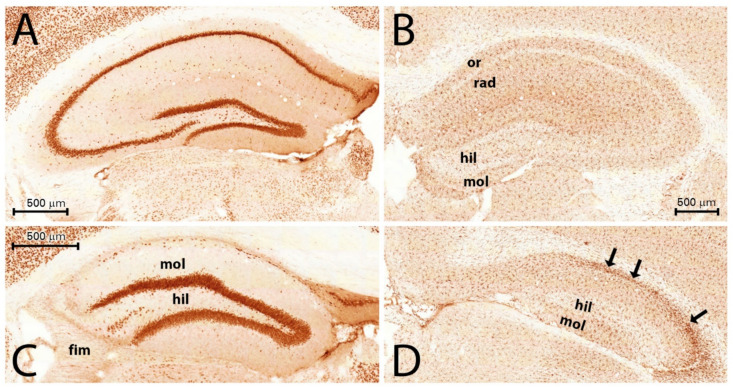
Neurons (**A**,**C**) and microglia (**B**,**D**) in control (**A**,**B**) and PT4 (**C**,**D**) hippocampi. NeuN staining (**A**,**C**) revealed complete loss of principal neurons in CA regions (PT4 animal (**C**)). Iba1 staining revealed the accumulation of microglia cells (arrows in (**D**) in the sclerosed regions). Abbreviations: mol = molecular layer of DG; hil = hilum of DG; or = stratum oriens; rad = stratum radiatum. Bars: 500 µm (bar on (**B**) is for (**D**)).

**Figure 2 ijms-23-00204-f002:**
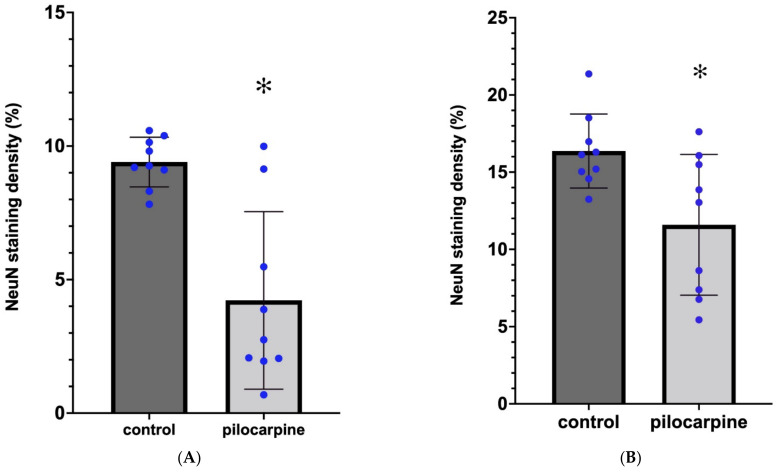
Densitometric analysis of NeuN-stained hippocampi (see Materials and Methods). (**A**) NeuN staining density in the entire Ammon’s horn in control and PT animals (n = 9; paired *t*-test, * *p* = 0.0021). (**B**) NeuN staining density in the entire area of the DG in control and PT animals (n = 9; Wilcoxon signed rank test, * *p* = 0.0039).

**Figure 3 ijms-23-00204-f003:**
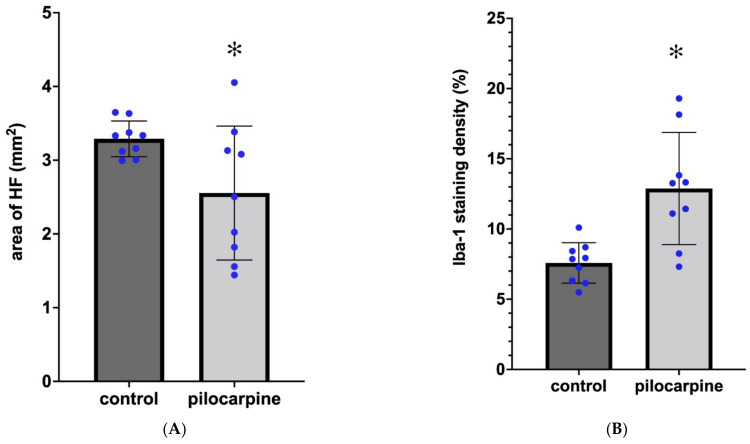
(**A**) Decrease in the cross-sectional area of the HF in PT animals compared with controls, as measured on coronal sections (n = 9; Wilcoxon signed rank test, * *p* = 0.0039). (**B**) Increase in the Iba1 staining density in the HF in PT animals compared with controls (n = 9; paired *t*-test, * *p* = 0.0111).

**Figure 4 ijms-23-00204-f004:**
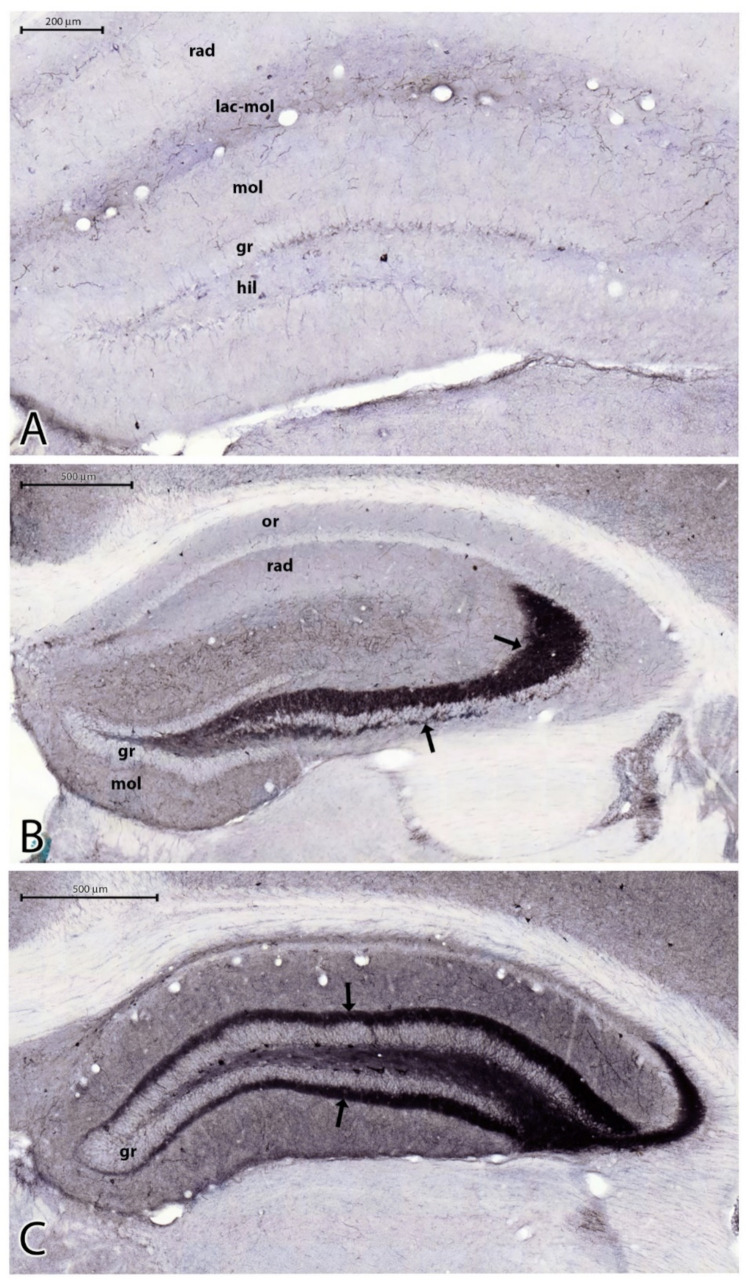
NPY immunostaining of control (**A**) and epileptic (**B**,**C**) hippocampi. (**B**) PT5 animal. (**C**) PT3 animal. Black staining indicates sprouting NPY-containing axons in the hilum, in stratum lucidum, and stratum oriens of CA3 ((**B**), arrows). The IML of the DG was also heavily infiltrated by sprouting axons ((**C**), arrows). Abbreviations: rad = stratum radiatum; lac-mol = stratum lacunosum-moleculare; mol = molecular layer of DG; gr = granular layer of DG; hil = hilum of DG. Bars: (**A**) 200 µm; (**B**,**C**) 500 µm.

**Figure 5 ijms-23-00204-f005:**
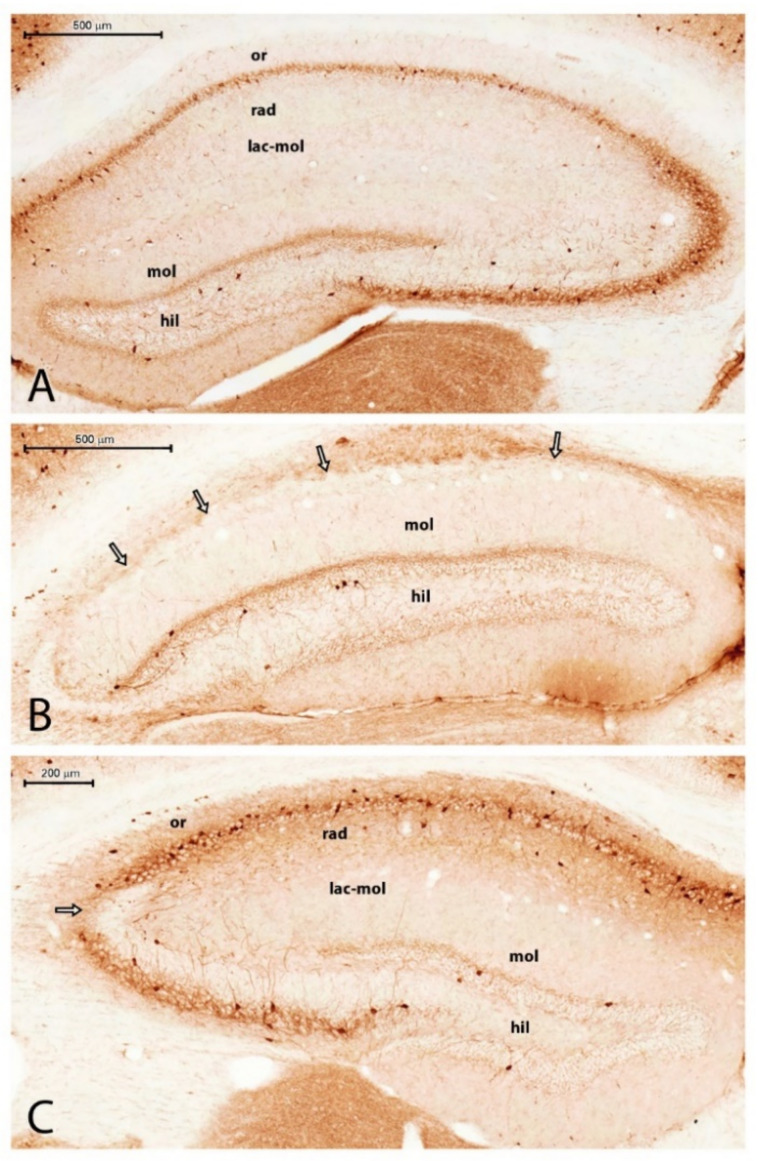
PV staining of the control (**A**) and epileptic (**B**,**C**) hippocampi. Note the scattered PV-stained neurons and pericellular PV-stained axonal network in the layers of the principal neurons in (**A**) (or = stratum oriens; rad = stratum radiatum; lac-mol = stratum lacunosum-moleculare; mol = molecular layer of DG; hil = hilum of DG; bar = 500 µm). (**B**) In animal PT7, there were no principal neurons in CA1–3, and the PV neurons disappeared, too (arrows point to the location of the stratum pyramidale without cells). Bar = 500 µm; mol = molecular layer of DG; hil = hilum of DG. (**C**) Patchy neuronal loss in CA3 (arrow). Abbreviations are the same as in (**A**,**B**); bar = 200 µm.

**Figure 6 ijms-23-00204-f006:**
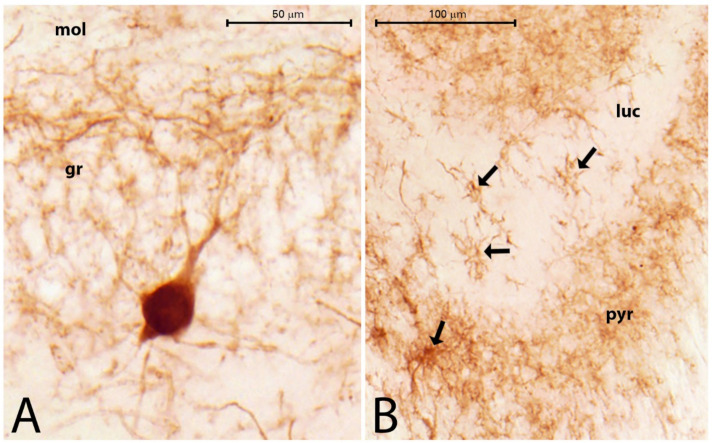
(**A**,**B**) PV-stained structures in PT4 (**A**) and PT7 (**B**), with a PV-stained neuron between the neurons of the DG granular layer (gr). The fine pericellular PV network is visible (mol = molecular layer of DG; bar = 50 µm). PV-stained astrocyte-like cells (black arrows) are visible in the stratum lucidum (luc) and stratum pyramidale (pyr) of CA3 in PT7 (bar = 100 µm).

**Figure 7 ijms-23-00204-f007:**
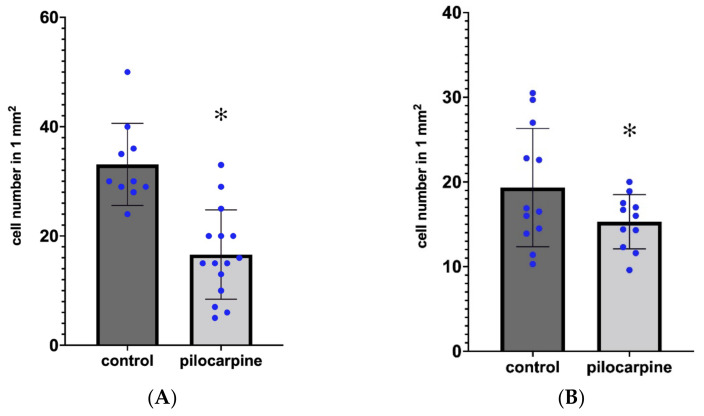
(**A**) The number of PV-containing neurons in the HF in the control and PT animals (n = 10 in control and n = 15 in PT; paired *t*-test, * *p* = 0.0005). (**B**) The number of CR-containing neurons in the HF in the control and PT animals (n = 12 in control, n = 11 in PT; Wilcoxon signed rank test, * *p* = 0.0005).

**Figure 8 ijms-23-00204-f008:**
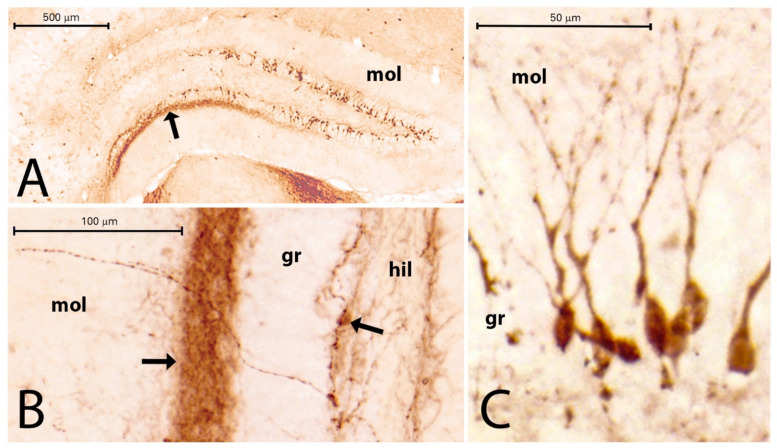
(**A**–**C**) CR staining in PT1 (**A**,**C**) and in PT8 (**B**). Note the numerous CR-IR granule cells in the DG of PT1 (**A**,**C**). The arrow in (**A**) points to the IML, where strong CR-IR cell processes are seen (mol = molecular layer of DG; bar = 500 µm). The CR-stained IML is seen with higher magnification in (**B**) (arrow). Note the scattered, varicose CR-IR fibers in the hilum (hil) and molecular layer (mol). The black arrow in the hilum in (**B**) points to the CR-IR neuronal cell body (gr = granular layer of DG). Bar = 100 µm.

**Figure 9 ijms-23-00204-f009:**
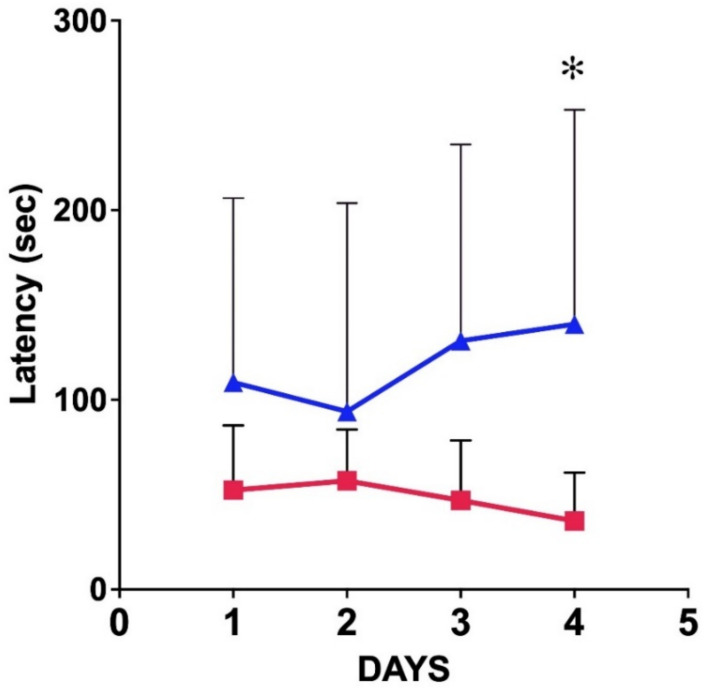
Performance of control (red) and PT (blue) animals in the Barnes maze. Mean latencies of the 9 control and 9 PT animals were plotted against time. The standard error of the mean is depicted at each point. It is clear that the controls showed improved performance as a function of time, while the performance of the PT animals became worse. The two-way ANOVA proved that the variances of the mean latencies in the groups were significantly different (asterisk: *p* = 0.0438; alpha = 0.05; 95% CI of the difference: from −138.5 to −2.209).

**Figure 10 ijms-23-00204-f010:**
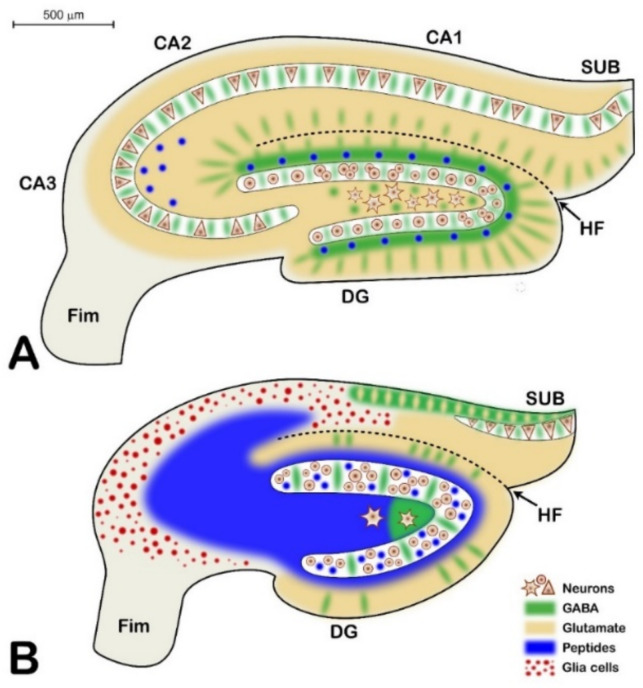
Scheme of the neurochemical parcellation of the hippocampus in the control (**A**) versus the PT (**B**) animals. Principal neurons are represented by triangular, circular, and multipolar shapes. GABAergic transmission is depicted in green, glutamate transmission is beige, and peptide transmitters are in blue. Glia is represented by red dots. The PT hippocampus is not only shrunken but also invaded by peptidergic axons, some of which do not have postsynaptic cells, and the peptide released is diffusing toward unknown targets. This neurochemical alteration will drastically change the function of the hippocampus, as shown by the Barnes maze experiments (CA1, CA2, and CA3 are the regions of the Ammon’s horn; DG = dentate gyrus; SUB = subiculum; Fim = fimbria; HF = hippocampal fissure; bar = 500 µm).

**Table 1 ijms-23-00204-t001:** The degree of HS and the localization of neuronal degeneration in PT mice. The fields of the Ammon’s horn (CA), hilum, and granule cell layer of the DG are indicated. The observations are based on the qualitative light microscopy of the NeuN immunostained histological sections of the HF.

Pilocarpine-Treated (PT) Mouse Number	Localization of the Light Microscopic Neuronal Loss in the Hippocampal Formation Stained with Anti-NeuN Serum
PT1	CA1, CA2, CA3 degenerated
PT2	only CA2 degenerated
PT3	CA1, CA2, CA3, hilum degenerated, granule cells dispersed
PT4	CA1, CA2, CA3 completely degenerated
PT5	cell loss in CA1, CA3 and hilum
PT6	cell loss in the hilum and upper blade of dentate granule cell layer
PT7	CA1, CA2 and CA3 completely degenerated, minor loss of the upper blade of the dentate granule cell layer was present
PT8	CA1, CA3: patchy neuronal losses in the stratum pyramidale
PT9	CA1, CA3 degenerated, hilum partial loss, CA2 normal

**Table 2 ijms-23-00204-t002:** The statistics of the latencies (in seconds) of finding the escape box in a Barnes maze on four consecutive days. Two trials were performed on each day, except day 4, which was the single probe trial. The statistical means of the latencies measured at the daily trials and the SDs are depicted. The number of animals was 9 in both groups. Using the Mann–Whitney test, the daily performances were tested statistically. A significant difference between the controls and PT animals was found only on day 4 (*p* = 0.0295 *, * denotes significant difference).

		Day 1	Day 2	Day 3	Day 4
**Control**	mean	52.3	57.2	46.9	36.1 *
	SD	34.1	27.1	31.7	25.4
**Pilo-treated**	mean	109.2	93.8	131.1	139.8 *
	SD	97.0	109.9	103.4	113.0

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
