# Peer review of "Hippocampal Sclerosis in Pilocarpine Epilepsy: Survival of Peptide-Containing Neurons and Learning and Memory Disturbances in the Adult NMRI Strain Mouse"

_ijms, 2021, doi:10.3390/ijms23010204_

Round 1
Reviewer 1 Report
It would be necessary to better specify the number of animals present in the control group and in the experimental group at the beginning of the protocol.
Citations in the results should only be included in discussions.
It might be interesting to relate these interesting findings to the clinic: for example, in one study, it was seen that children with epilepsy can have visuospatial memory deficits compared to their peers and that different AEDs can affect visuospatial memory differently, therefore it is important to monitor this aspect in pediatric patients, also in light of the results presented in the article.
As a matter of logical sequencing, it would be advisable to describe the methods before the results and not after the discussions.
Reviewer 2 Report
Line 68-73: The authors may want to raise the specific research questions explicitly. It is unclear why it matters to describe such epilepsy in such animal model. Is there any unsolved questions that the authors are trying to answer?
Line 88: Does "PILO-" stand for "pilocarpine"? The author may want to define the abbreviation explicitly before using it.
Line 97: what are the behaviors and what results when the authors say "the behaviour of the seizing animals was compared to healthy controls"?
Figure 1: The authors may want to refer to Figure 9 for a diagram showing the locations of CA1, CA2 and CA3.
Figure 3: Why use Wilcoxon signed rank test in 3A while using paired t-test in 3B?
Figure 6: Why is n=10 in 6A control but n=15 in 6A PT? What are the selection criteria of data points used to generate the plot? Why use paired t-test in 6A while Wilconxon signed rank test in 6B?
Line 254: Why using Mann Whitney test instead of t test?
